# Enhanced Multi-Stream Remote Sensing Spatiotemporal Fusion Network Based on Transformer and Dilated Convolution

**Weisheng Li** *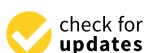, **Dongwen Cao** and **Minghao Xiang**

College of Computer Science and Technology, Chongqing University of Posts and Telecommunications, Chongqing 400065, China
* Correspondence: liws@cqupt.edu.cn

**Abstract:** Remote sensing images with high temporal and spatial resolutions play a crucial role in land surface-change monitoring, vegetation monitoring, and natural disaster mapping. However, existing technical conditions and cost constraints make it very difficult to directly obtain remote sensing images with high temporal and spatial resolution. Consequently, spatiotemporal fusion technology for remote sensing images has attracted considerable attention. In recent years, deep learning-based fusion methods have been developed. In this study, to improve the accuracy and robustness of deep learning models and better extract the spatiotemporal information of remote sensing images, the existing multi-stream remote sensing spatiotemporal fusion network MSNet is improved using dilated convolution and an improved transformer encoder to develop an enhanced version called EMSNet. Dilated convolution is used to extract time information and reduce parameters. The improved transformer encoder is improved to further adapt to image-fusion technology and effectively extract spatiotemporal information. A new weight strategy is used for fusion that substantially improves the prediction accuracy of the model, image quality, and fusion effect. The superiority of the proposed approach is confirmed by comparing it with six representative spatiotemporal fusion algorithms on three disparate datasets. Compared with MSNet, EMSNet improved SSIM by 15.3% on the CIA dataset, ERGAS by 92.1% on the LGC dataset, and RMSE by 92.9% on the AHB dataset.

**Keywords:** spatiotemporal fusion; dilated convolution; improved transformer encoder; global correlation information



## 1. Introduction

Remote sensing images are generated by various types of satellite sensors, such as the Moderate Resolution Imaging Spectroradiometer (MODIS), Landsat-equipped sensors, and Sentinel. MODIS sensors are usually installed on Terra and Aqua satellites, which can circle the earth in half a day or one day, and the data obtained by them have superior time resolution. However, the spatial resolution of MODIS data (i.e., rough image) is very low, and accuracy can reach only 250–1000 m [1]. By contrast, data (fine image) acquired by Landsat have higher spatial resolution (15–30 m) and capture sufficient surface-detail information, but temporal resolution is very low because it takes 16 days to circle the earth [1]. In practical applications, we often need remote sensing images with high temporal and spatial resolution. For example, images with high temporal and spatial resolutions can be used for research in the fields of heterogeneous regional surface change [2,3], vegetation seasonal monitoring [4], real-time natural disaster mapping [5], and land-cover changes [6]. Unfortunately, current technical and cost constraints, coupled with the existence of such noise as cloud cover in some areas, make it challenging to directly obtain remote sensing products with high temporal and spatial resolution, and a single high-resolution image cannot meet practical needs. In order to meet these lacunae, spatiotemporal fusion has attracted considerable attention. In spatiotemporal fusion, two types of images are fused together, with the aim of obtaining images with high spatiotemporal resolution [7,8].

Existing spatiotemporal fusion methods can generally be subdivided into four categories: unmixing-based, reconstruction-based, dictionary pair learning-based, and deep learning-based.

Unmixing-based methods unmix the spectral information at the predicted moment, and then use the unmixed result to predict the unknown high spatial and temporal resolution image. Multi-sensor multi-resolution image fusion (MMFN) [9] was the first fusion method to apply the idea of unmixing. MMFN reconstructs the MODIS and Landsat images separately: first, the MODIS image is spectrally unmixed, and then the mixed result is spectrally reset on the Landsat image to obtain the final reconstruction result. Wu et al. considered the issue of nonlinear time-varying similarity and spatial variation in spectral unmixing, improved MMFN, and obtained a new spatiotemporal fusion method, STDFA [10], which also achieved good fusion results. A variable spatiotemporal data-fusion algorithm, FSDAF [11], has also been proposed, which combines the unmixing method, spatial interpolation, and spatiotemporal adaptive fusion algorithm (STARFM) to create a new algorithm that is computationally inexpensive, fast, and accurate, and performs well in heterogeneous regions.

The core idea of the reconstruction-based algorithm is to calculate the weights of similar adjacent pixels in the spectral information in the input and then add them. STARFM was the first method to be used for reconstruction for fusion [8]. In STARFM, the reflection changes of pixels between the rough image and the fine image should be continuous, and the weights of adjacent pixels can be calculated to reconstruct a surface-reflection image with high spatial resolution. In light of STARFM's large number of computations and the need to improve the reconstruction effect for heterogeneous regions, Zhu et al. made improvements and proposed an enhanced version of STARFM called ESTARFM [12]. They use two different coefficients to deal with the weights of adjacent pixels in homogeneous and heterogeneous regions, achieving a better effect. Inspired by STARFM, the spatiotemporal adaptive algorithm for mapping reflection changes (STAARCH) [13] also achieves good results. Overall, the difference between these algorithms lies in how the weights of adjacent pixels are calculated. Although these algorithms generally have good results, they are unsuitable for data that change too much too quickly.

Dictionary learning-based methods mainly learn the correspondence between two types of remote sensing images to perform prediction. The sparse representation-based spatiotemporal reflection fusion method (SPSTFM) [14] may be the first fusion method to successfully apply dictionary learning. In SPSTFM, the coefficients of low-resolution images and high-resolution images should be the same, and the super-resolution ideas in the field of natural images are introduced into spatiotemporal fusion. Images are reconstructed by establishing correspondences between low-resolution images. However, in practical situations, the same coefficients may not be applicable to some of the data obtained under the existing conditions [15]. Wei et al. studied the explicit mapping between low-resolution images and proposed a new fusion method based on dictionary learning and utilizing compressive sensing theory, called compressive sensing spatiotemporal fusion (CSSF) [16], which improves the accuracy of the prediction results noticeably, but the training time also increases considerably, while the efficiency decreases. In this regard, Liu et al. proposed an extreme learning machine called ELM-FM for spatiotemporal fusion [17], which considerably reduces time and improves efficiency.

As deep learning has gradually been applied in various fields in recent years, deep learning-based spatiotemporal fusion methods of remote sensing have also advanced. For example, Song et al. proposed STFDCNN [18] for spatiotemporal fusion using a convolutional neural network. In STFDCNN, the image-reconstruction process is considered a super-resolution and nonlinear mapping problem. A super-resolution network and a nonlinear mapping network are constructed through an intermediate resolution image, and the final fusion result is obtained through high-pass modulation. STFDCNN achieved good results. Liu et al. proposed a two-stream CNN, StfNet [19], for spatiotemporal fusion. They effectively extracted and fused spatial details and temporal information using spatial consistency and temporal dependence, and achieved good results. On the basis of spatial consistency and time dependence, Chen et al. introduced a multiscale mechanism for

feature extraction and proposed a spatiotemporal remote sensing image-fusion method based on multiscale two-stream CNN (STFMCNN) [20]. Jia et al. proposed a new deep learning-based two-stream convolutional neural network [21], which fuses the temporal variation information with the spatial detail information by weight, which enhances its robustness. Furthermore, Jia et al. adopted various prediction methods for phenological change and land-cover change, and proposed a spatiotemporal fusion method based on hybrid deep learning to combine satellite images with differing resolutions [22]. Tan et al. proposed DCSTFN [23] to derive high spatiotemporal remote sensing images using CNNs based on the methods of convolution and deconvolution combined with the fusion method of STARFM. However, in light of the loss of information in the reconstruction process of the deconvolution fusion method, Tan et al. increased the input of the a priori moment and added a residual coding block, using a composite loss function to improve the learning ability of the network, and an enhanced convolutional neural network EDCSTFN [24] was proposed for spatiotemporal fusion. In addition, CycleGAN-STF [25] introduces other ideas in the visual field into spatiotemporal fusion. It achieves spatiotemporal fusion through image generation of CycleGAN. CycleGAN is used to generate a fine image at the predicted time, the real image is used at the predicted time to select the closest generated image, and finally FSDAF is used for fusion. Other fusion methods are applied in specific scenarios. For example, STTFN [26], a CNN-based model for spatiotemporal fusion of surface-temperature changes, uses a multiscale CNN to establish a nonlinear mapping relationship and a spatiotemporal continuity weight strategy for fusion, achieving good results. DenseSTF [27], a deep learning-based spatiotemporal data-fusion algorithm, uses a block-to-point modeling strategy and model comparison to provide rich texture details for each target pixel to deal with heterogeneous regions, and achieves very good results. Furthermore, with the development of transformer models [28] in the natural language field, many researchers have introduced the concept into the vision field as well, e.g., vision transformer (ViT) [29], data-efficient image transformer (DeiT) [30], conditional position encoding visual transformer (CPVT) [31], transformer-in-transformer (TNT) [32], and convolutional vision transformer (CvT) [33] can be used for image classification. In addition, there are the Swin transformer [34] for image classification, image segmentation, and object detection, and texture transformer [35] for general image superclassification. These variants have been gradually introduced into the spatiotemporal fusion of remote sensing. For example, MSNet [36] is a new method obtained by introducing the original transformer and ViT into spatiotemporal fusion, learning the global temporal correlation information of the image through the transformer structure, using the convolutional neural network to establish the relationship between input and output, and finally obtain a good effect. SwinSTFM [37] is a new method that introduces the Swin transformer and combines linear spectral mixing theory, which finally improves the quality of generated images. There is also MSFusion [38], which introduces texture transformer into spatiotemporal fusion, which has also achieved quite good results on multiple datasets.

Existing spatiotemporal fusion algorithms perform a certain amount of information extraction and noise processing during the fusion process, but there remain certain lacunae. First, the acquisition and processing of suitable datasets is not easy. Owing to the existence of noise, the data that can be directly used for research are insufficient. In deep learning, the size of the dataset affects the learning ability during reconstruction: achieving good reconstruction with small datasets is a major challenge. Second, the same fusion model can have different prediction performance on different datasets, and the model is not robust. Furthermore, the features extracted only by the CNN are not sufficient, and an increase of the network depth will also result in potential feature loss.

In order to address the aforementioned challenges, this study improves MSNet and proposes an enhanced version of the spatiotemporal fusion method of multi-stream remote sensing images called EMSNet. In EMSNet, the input image adopts the original scale size, and the rough image is no longer scaled to fully extract the temporal information and reduce the loss. The main contributions of this paper are summarized as follows.

(1) The number of prior input images required by the model is reduced from five to three, which achieves better results with less input, so that even a dataset with a small amount of data can reconstruct images with better effects.

(2) The transformer encoder structure is introduced and its projection method improved to obtain the improved transformer encoder (ITE), which adapts the remote sensing spatiotemporal fusion, effectively learns the relationship between local and global information in rough and fine images, and effectively extracts temporal and spatial information.

(3) Dilated convolution is used to extract temporal information, which expands the receptive field while keeping the parameter quantity unchanged and fully extracts a large amount of temporal feature information contained in the rough image.

(4) A new feature-fusion strategy is used to fuse the features extracted by the ITE and dilated convolution based on their differences from real predicted images in order to avoid introducing noise.

The rest of the article has the following structure. The overall structure of EMSNet and its internal specific modules and weight strategies are introduced in Section 2. Experimental results are described in Section 3, along with the datasets used. Section 4 dis-cusses the performance of EMSNet. Finally, conclusions are provided.

## 2. Methods

### 2.1. EMSNet Architecture

Figure 1 shows the overall structure of EMSNet, where $M_i(i = 1, 2)$ represents the MODIS image at time $t_i$, $L_i$ represents the Landsat image at time $t_i$, and $Pre\_L_2$ represents the prediction result of the fused image at time $t_2$ based on time $t_1$. Rectangles of different colors represent different operations, including convolution, dilated convolution, activation function ReLU, and various operations inside the improved transformer encoder (ITE). EMSNet is an end-to-end structure, which can be divided into three parts:

a. ITE-related modules, used to extract temporal change information and spatial texture detail features and learn local and global correlation information;

b. an extraction network composed of convolution and dilated convolution, used to establish a nonlinear relationship between input and output, while fully extracting the features of time information;

c. a weight strategy, used to calculate the corresponding weight according to the difference between the features obtained in the above two parts and the real prediction map for final fusion.

A detailed description of each module can be found in Sections 2.2–2.4.

In this study, three images of the same size are used as input, a pair of MODIS-Landsat images at a priori time $t_1$ and a MODIS image at prediction time $t_2$. The overall procedure of EMSNet is as follows:

(1) First, we subtract $M_1$ from $M_2$ to get $M_{12}$, which represents the change area within two times and provides time-change information. We input into the feature-extraction network composed of convolution and dilated convolution, and then fully extract the time information contained in it.

(2) Second, we add $M_{12}$ and $L_1$ to the ITE to extract the rich temporal information and spatial texture detail information, and simultaneously learn the connection between the local and the global information.

(3) Inspired by ResNet [39], in DenseNet [40], as the network depth increases, the temporal and spatial information in the input image may be lost during transmission. Therefore, we add $L_1$ as the residual to the temporal variation information obtained in the first step to supplement the spatial details that may be lost in the subsequent fusion process.

(4) Finally, the results obtained in the second and third steps are calculated by calculating the difference with $L_2$ to obtain their respective weights, so as to fuse and reconstruct the final prediction map $Pre\_L_2$.

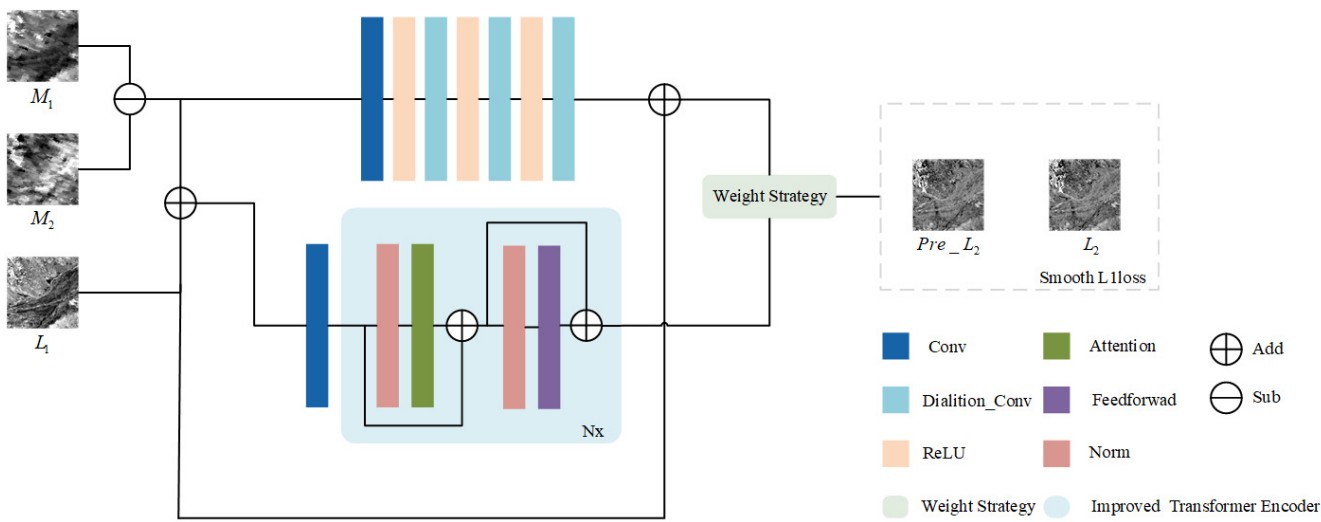

**Figure 1.** EMSNet architecture.

The structure of EMSNet can be represented by Equation (1) below:

$$Pre\_L_2 = W(T(M_{12} + L_1), E(M_{12}) + L_1) \tag{1}$$

Here, $T$ represents the ITE module, $E$ represents the time information-extraction network composed of convolution and dilated convolution, and $W$ represents the weight strategy adopted in this study.

### 2.2. Improved Transformer Encoder

Transformer [28], as a kind of attention mechanism, is well suited not only to the field of natural language but also to the field of vision. Inspired by the application of the transformer in MSNet [36] and the cancellation of position encoding in CPVT [31] and CvT [33], in this study, the transformer encoder applied to remote sensing spatiotemporal fusion is further improved, the MLP part for classification is canceled, and position encoding is canceled. In addition, the convolutional projection method is used to replace the original linear projection method in the transformer, and a new structure, as shown in Figure 2 below, is obtained, called the improved transformer encoder (ITE), which is mainly used to learn temporal variation information and spatial texture details. Through the above operations, it is ensured that the input and output are of the same dimension, which facilitates subsequent fusion and reconstruction.

Figure 2 is the ITE structure diagram, in which the yellow part represents the convolution projection operation and the blue box and its interior represent the specific operation part of ITE. As can be seen from the figure, this study projects the input information directly through the convolution operation, and the overlap between the convolution blocks and the convolution blocks effectively strengthens the connection between the blocks. Consequently, the ITE strengthens the correlation between local information and global information, removing the need for the position encoding required by the linear projection method, thus making it more suitable for the spatiotemporal fusion method. The ITE is also composed of alternate multi-head attention mechanisms and feedforward parts. It will be normalized before each input to the submodule, and there will be residual connections after each block. The multi-head self-attention mechanism is a series of SoftMax and linear operations, and the input data will gradually change the dimensions during the propagation and training process to adapt to match these operations. The feedforward portion is composed of linear, Gaussian error linear unit (GELU), and random deactivation dropout, where GELU is used as the activation function. In practical applications, for different amounts of data, when learning global time-varying information, ICTE with different depths are required to learn more accurately. Nx in the figure represents the depth value.

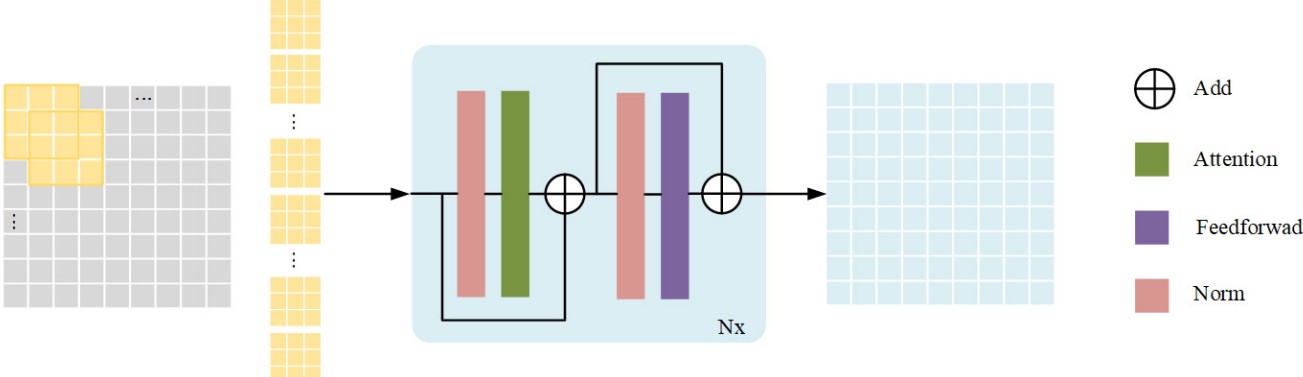

**Figure 2.** Structure of improved transformer encoder (ITE).

In this study, the ITE is used as a module for learning time-varying information and spatial texture detail. Compared with the previous MSNet, it further expands the learning range of the transformer encoder for remote sensing.

### 2.3. Dilated Convolution

In order to extract the time information contained in $M_{12}$ and establish the mapping relationship between input and output, this study proposes a seven-layer neural network mainly composed of dilated convolution as a feature extraction network. The key feature of dilated convolution is that different sizes of receptive fields can be obtained after setting different dilation rates, so as to extract effective information at multiple scales. Compared with ordinary convolution operations, dilated convolution will not increase the number of redundant parameters. Figure 3 shows the proposed dilated convolution-based neural network and the receptive fields under different dilation rates.

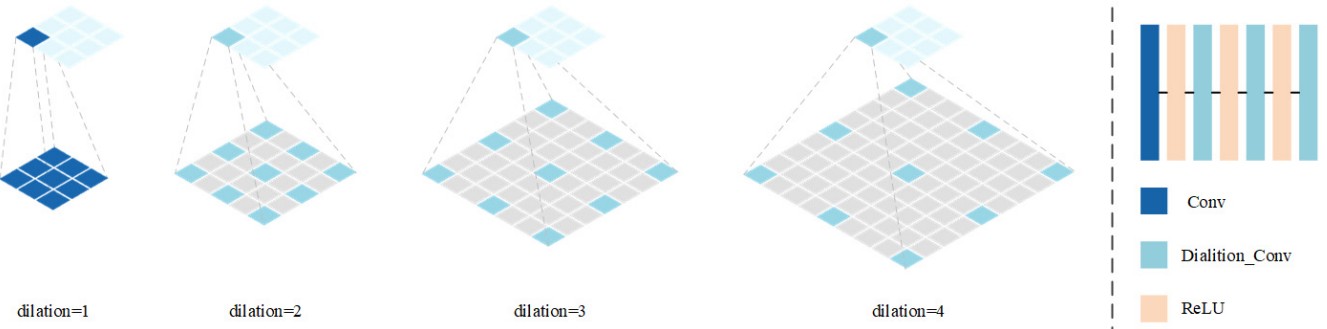

dilation=1     dilation=2     dilation=3     dilation=4

Conv
Dialition_Conv
ReLU

**Figure 3.** Neural network based on dilated convolution and receptive fields with different dilation rates.

The right side of the dotted line in Figure 3 shows the architecture of the seven-layer neural network, which consists of one layer of convolution, three layers of dilated convolution, and three layers of ReLU. The convolution operation is used to convert the original $M_{12}$ into a multidimensional nonlinear tensor, and the convolution kernel adopts the size of $3 \times 3$; the dilated convolution is used to effectively extract the temporal features in the $M_{12}$, the basic convolution kernel is of the same size i.e., $3 \times 3$, and an expansion rate of 2, 3, and 4 is set in turn for three consecutive layers of dilated convolution. The left side of the dotted line is the schematic diagram of the receptive field under various expansion rates. When the dilation rate is 1, dilated convolution is no different from ordinary convolution. When the dilation rate increases, the receptive field also gradually increases, which enables it to better learn the feature information at various scales, and simultaneously guarantee the number of parameters taken during its operation will not increase [41].

Each dilated convolution operation can be defined as:

$$\Phi(x) = w_i * x + b_i \tag{2}$$

Here, $x$ represents the input, "$*$" represents the dilated convolution operation, $w_i$ represents the weight of the current convolutional layer, and $b_i$ represents the current offset. The output channels of the three convolution operations are 32, 16, and 1 in sequence. After the convolution, the ReLU operation is used to make the features non-linear and avoid network overfitting [42]. The ReLU operation can be defined as:

$$\text{ReLU}(x) = \max(0, x) \tag{3}$$

*2.4. Weight Strategy*

After feature extraction by the ITE and dilated convolutional neural network, plus residual $L_2$ for supplementary information, two distinct features are obtained. The difference between the prediction graphs is calculated by weight for final fusion, and the specific weight strategy can be defined as:

$$Pre\_L_2 = W(T(M_{12} + L_1), E(M_{12}) + L_1) = \alpha T(M_{12} + L_1) + \beta(E(M_{12}) + L_1) \tag{4}$$

$$\begin{cases} \alpha = \dfrac{\frac{1}{|T(M_{12}+L_1)-L_2|}}{\frac{1}{|T(M_{12}+L_1)-L_2|} + \frac{1}{|(E(M_{12})+L_1)-L_2|}} \\[4mm] \beta = \dfrac{\frac{1}{|(E(M_{12})+L_1)-L_2|}}{\frac{1}{|T(M_{12}+L_1)-L_2|} + \frac{1}{|(E(M_{12})+L_1)-L_2|}} \end{cases} \tag{5}$$

Here, $T$ represents the ITE module, and $E$ represents the temporal information extraction network composed of convolution and dilated convolution.

*2.5. Network Training*

During the entire training process of the model, the loss calculation is performed on the prediction results of the entire model, so as to continuously adjust the learning parameters during the backpropagation process to obtain better convergence results. When calculating the difference between the predicted result and the real value, the smooth L1 loss function, namely Huber loss [43], is chosen, which can be defined as $\mathcal{L}$:

$$S(L_i) = \sum_{m=1}^{H} \sum_{n=1}^{W} L_i(m, n) \tag{6}$$

$$\mathcal{L} = loss(Pre\_L_2, L_2) = \frac{1}{N} \begin{cases} \frac{1}{2}(S(Pre\_L_2) - S(L_2))^2, & if\,|S(Pre\_L_2) - S(L_2)| < 1 \\[2mm] |S(Pre\_L_2) - S(L_2)| - \frac{1}{2}, & otherwise \end{cases} \tag{7}$$

where $H$ represents the height of the image, $W$ represents the width of the image, $L_i$ represents the input image, and $S$ represents for the pixel sum formula.

## 3. Experiments and Results

### 3.1. Datasets

Three separate datasets were employed to test the robustness of EMSNet.

The first study area was the Coleambally Irrigation Area (CIA) in southern New South Wales (NSW, Australia, 34.0034°E, 145.0675°S) [44]. The dataset was acquired from October 2001 to May 2002 and comprises 17 pairs of MODIS–Landsat images. The Landsat images are all from Landsat-7 ETM+, and the MODIS images are MODIS Terra MOD09GA Collection 5 data. The CIA dataset includes six bands and an image size of 1720 × 2040.

The second study area is the Lower Gwydir Watershed (LGC) in northern New South Wales (NSW, 149.2815°E, 29.0855°S), Australia [44]. The dataset was acquired from

April 2004 to April 2005 and comprises 14 pairs of MODIS–Landsat images. All Landsat imagery is from Landsat-5$^{TM}$, and the MODIS imagery is MODIS Terra MOD09GA Collection 5 data. The LGC dataset contains six bands and the image size is 3200 × 2720.

The third study area is the Alu Horqin Banner (AHB) region (43.3619°N, 119.0375°E) in the central Inner Mongolia Autonomous Region of northeastern China, which has many circular pastures and farmland [45,46]. Li Jun et al., collected 27 cloud-free MODIS–Landsat image pairs from 30 May 2013 to 6 December 2018, a time span of more than 5 years. The area has experienced substantial phenological changes owing to the growth of crops and other types of vegetation. The AHB dataset contains six bands and the image size is 2480 × 2800.

In this study, all images of the three datasets are combined according to a prior time and a prediction time. Each set of training data has four images, including two pairs of MODIS–Landsat images. The image size of each pair of MODIS-Landsat is the same, and the spatial resolution is 16:1. When combining the data, the data with the same time span between the prior moment and the predicted moment are given priority as the experimental data. In addition, for the training of the network, the images of the three datasets are all adjusted to a size of 1200 × 1200. Figures 4–6 show the MODIS–Landsat image pairs obtained on two different dates for the three datasets. During the experiment process, the three datasets were input into EMSNet for training, 70% of the dataset was used for training, 15% was used for validation, and 15% was used as the final test set for evaluating the fusion and reconstruction ability of the model.

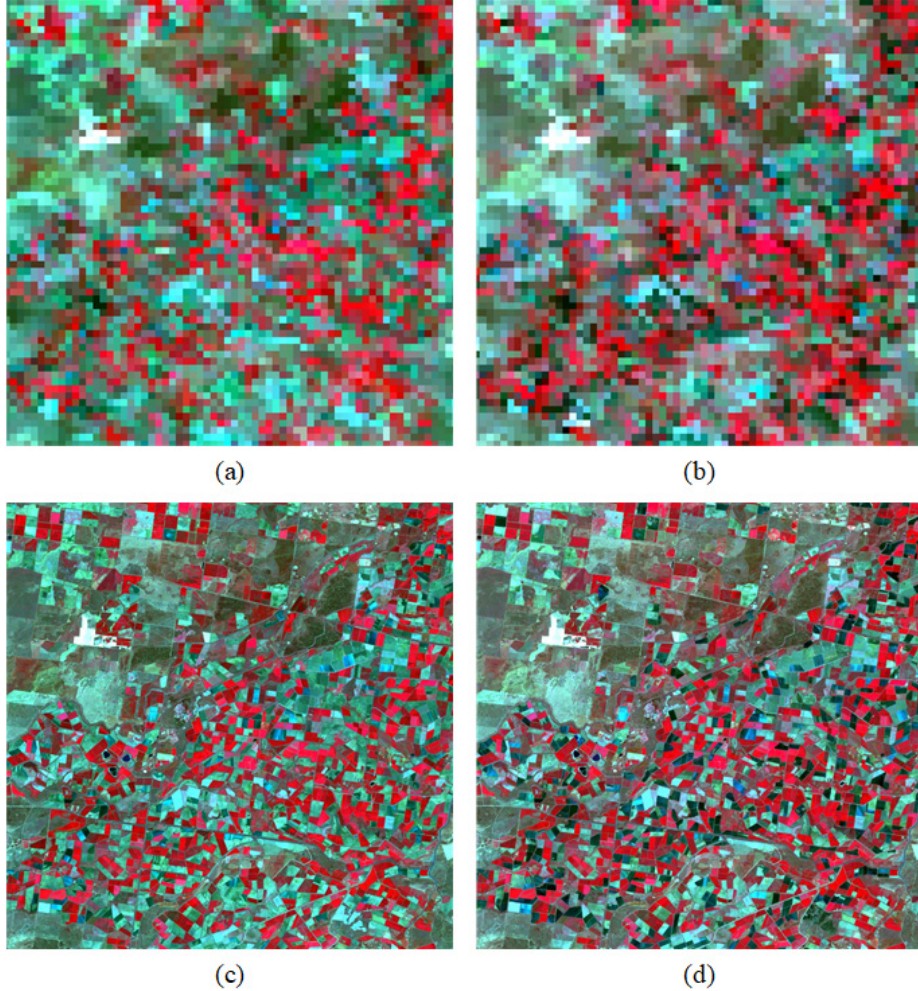

(a) (b)

(c) (d)

**Figure 4.** Composite MODIS (**Top row**) and Landsat (**Bottom row**) image pairs on 7 October (**a**,**c**) and 16 October (**b**,**d**) 2001 on the CIA [44] dataset. The CIA dataset focuses on noteworthy phenological changes in irrigated farmland.

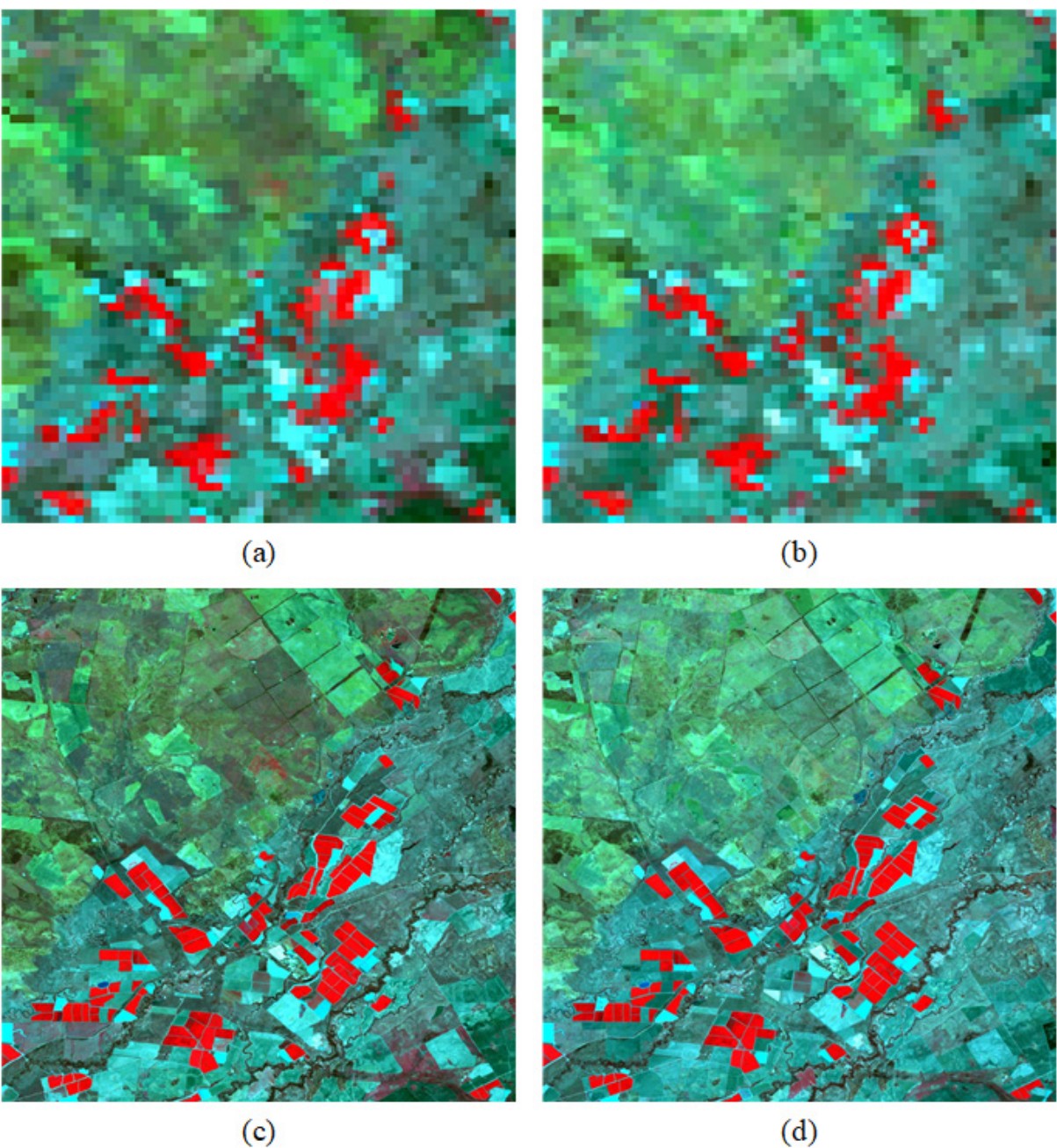

**Figure 5.** Composite MODIS (**top row**) and Landsat (**bottom row**) image pairs on 29 January (**a,c**) and 14 February (**b,d**) 2005 from the LGC [44] dataset. The LGC dataset focuses on changes in land cover types after the flood.

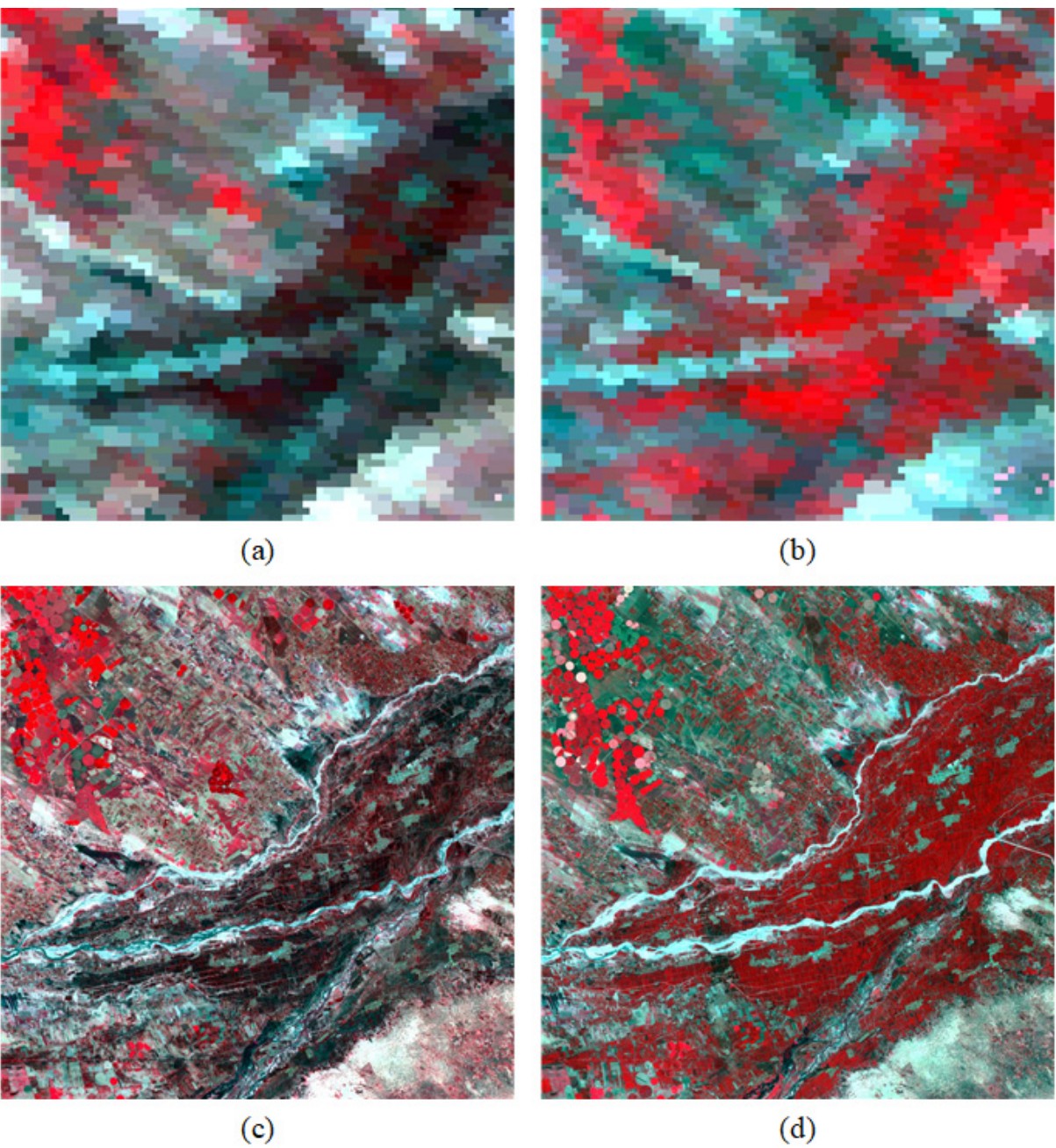

**Figure 6.** Composite MODIS (**top row**) and Landsat (**bottom row**) image pairs on 21 June (**a**,**c**) and 7 July (**b**,**d**) 2015 from the AHB [45,46] dataset. The AHB focuses on noteworthy phenological changes in the pasture.

### 3.2. Evaluation

We evaluated the proposed spatiotemporal fusion method by comparing it with FSDAF, STARFM, DCSTFN, STFDCNN, StfNet, and the previous MSNet under the same criteria.

As in the case of MSNet, six evaluation metrics are used. The first indicator is the spectral angle mapper (SAM) [47], which can measure the spectral distortion of the fusion result. It can be defined as follows:

$$\text{SAM} = \frac{1}{N} \sum_{n=1}^{N} \arccos \frac{\sum_{j=1}^{K} = (L_i^k Pre\_L_i^k)}{\sqrt{\sum_{j=1}^{K} (L_i^k)^2 \sum_{j=1}^{K} (Pre\_L_i^k)^2}} \tag{8}$$

where $N$ represents the total number of pixels in the predicted image, $K$ represents the total number of bands, $Pre\_L_i$ represents the prediction result, $Pre\_L_i^k$ represents the prediction result of the $k$th band, and $L_i^k$ represents the true value of the $L_i^k$ band. A small SAM indicates a better result.

The second metric was the root mean square error (RMSE), which is the square root of the MSE and is used to measure the deviation between the predicted image and the observed image. It reflects a global depiction of the radiometric differences between the fusion result and the real observation image, which is defined as follows:

$$\text{RMSE} = \sqrt{\frac{\sum\limits_{m=1}^{H}\sum\limits_{n=1}^{W}(L_i(m,n) - Pre\_L_i(m,n))^2}{H \times W}} \tag{9}$$

where $H$ represents the height of the image, $W$ represents the width of the image, $L$ represents the observed image, and $Pre\_L_i$ represents the predicted image. The smaller the value of RMSE, the closer the predicted image is to the observed image.

The third indicator was erreur relative global adimensionnelle de synthèse (ERGAS) [48], which measures the overall integration result. It can be defined as:

$$\text{ERGAS} = 100\frac{h}{l}\sqrt{\frac{1}{K}\sum\limits_{i=1}^{K}[\text{RMSE}(L_i^k)^2/(\mu_k)^2]} \tag{10}$$

where $h$ and $l$ represent the spatial resolution of Landsat and MODIS images respectively; $L_i^k$ represents the real image of the $k$th band; and $\mu_k$ represents the average value of the $k$th band image. When ERGAS is small, the fusion effect is better.

The fourth index was the structural similarity (SSIM) index [49], which is used to measure the similarity of two images. It can be defined as:

$$\text{SSIM} = \frac{(2\mu_{Pre\_L_i}\mu_{L_i} + c_1)(2\sigma_{Pre\_L_iL_i} + c_2)}{(\mu_{Pre\_L_i}^2 + \mu_{L_i}^2 + c_1)(\sigma_{Pre\_L_i}^2 + \sigma_{L_i}^2 + c_2)} \tag{11}$$

where $\mu_{Pre\_L_i}$ represents the mean value of the predicted image, $\mu_{L_i}$ represents the mean value of the real observation image, $\sigma_{Pre\_L_iL_i}$ represents the covariance of the predicted image $Pre\_L_i$ and the real observation image $L_i$, $\sigma_{Pre\_L_i}^2$ represents the variance of the predicted image $Pre\_L_i$, $\sigma_{L_i}^2$ represents the variance of the real observation image $L_i$, and $c_1$ and $c_2$ are constants used to maintain stability. The value range of SSIM is $[-1, 1]$. The closer the value is to 1, the more similar are the predicted image and the observed image.

The fifth index is the correlation coefficient (CC), which is used to indicate the correlation between two images. It can be defined as:

$$\text{CC} = \frac{\sum\limits_{n=1}^{N}(Pre\_L_i^n - \mu_{\hat{L}_i})(L_i^n - \mu_{L_i})}{\sqrt{\sum\limits_{n=1}^{N}(Pre\_L_i^n - \mu_{\hat{L}_i})^2}\sqrt{\sum\limits_{n=1}^{N}(L_i^n - \mu_{L_i})^2}} \tag{12}$$

The closer the CC is to 1, the greater the correlation between the predicted image and the real observation image.

The sixth indicator is the peak signal-to-noise ratio (PSNR) [50]. It is defined indirectly by the MSE, which can be defined as:

$$\text{MSE} = \frac{1}{HW}\sum\limits_{m=1}^{H}\sum\limits_{n=1}^{W}(L_i(m,n) - Pre\_L_i(m,n))^2 \tag{13}$$

Then PSNR can be defined as:

$$\text{PSNR} = 10 \cdot \log_{10}\left(\frac{MAX_{L_i}^2}{\text{MSE}}\right) \tag{14}$$

where $MAX_{L_i}^2$ is the maximum possible pixel value of the real observation image $L_i$. If each pixel is represented by an 8-bit binary value, then $MAX_{L_i}$ is 255. Generally, if the pixel value is represented by B-bit binary, then $MAX_{L_i} = 2^B - 1$. PSNR can evaluate the quality of the image after reconstruction. A higher PSNR means that the predicted image quality is better.

### 3.3. Parameter Settings

For the improved transformer encoder, the number of heads is set to 9, and the depth is set according to the data volume and characteristics of the three datasets: CIA is 20, LGC is 5, and AHB is 20. The size of the patch input into it is 240 × 240. The ordinary convolution as well as the three-layer dilated convolution in the dilated convolutional neural network each use a 3 × 3 convolution kernel. The dilation rates are 2, 3, and 4, and the number of channels is 32, 16, and 1. The initial learning rate is set to 0.0008, the optimizer adopts Adam, and the weight decay is set to $1 \times 10^{-6}$. EMSNet was trained on two Windows 10 Professional editions, each with 64 GB memory, an Intel Core i9-9900K @ 3.60 GHz×16 CPU, and an NVIDIA Geforce RTX 2080 Ti.

### 3.3.1. Subjective Evaluation

In order to visualize the experimental results, Figures 7–13 show the experimental results of FSDAF, STARFM, DCSTFN, STFDCNN, StfNet, MSNet, and the proposed improved EMSNet on each of three datasets. GT in the figure represents the real observed image, while Proposed is the proposed EMSNet method.

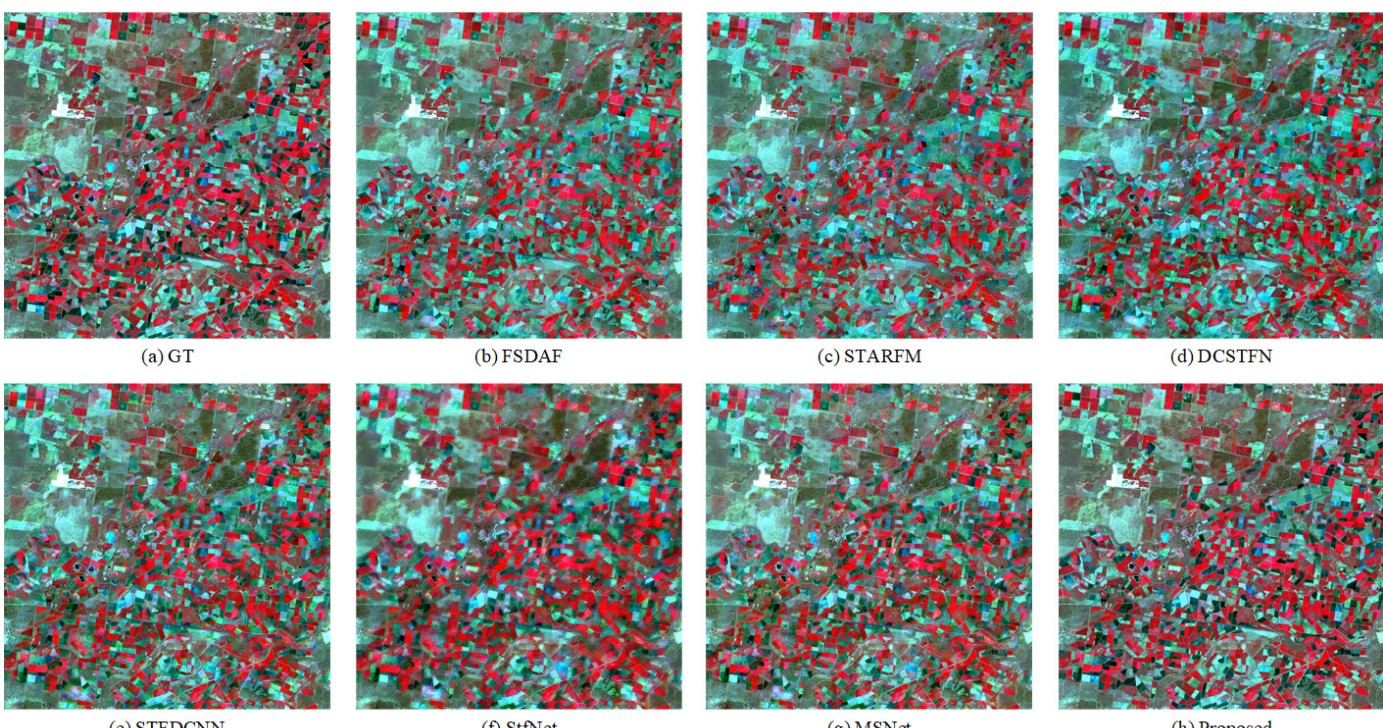

(a) GT     (b) FSDAF     (c) STARFM     (d) DCSTFN

(e) STFDCNN     (f) StfNet     (g) MSNet     (h) Proposed

**Figure 7.** Entire prediction results for the target Landsat image (16 October 2001) in the CIA dataset. Comparison methods include FSDAF [11], STARFM [8], DCSTFN [23], STFDCNN [18], StfNet [19], and MSNet [36], which are represented by (**b**–**g**) in the figure respectively. (**a**) represents the ground truth (GT), and (**h**) represents the proposed method.

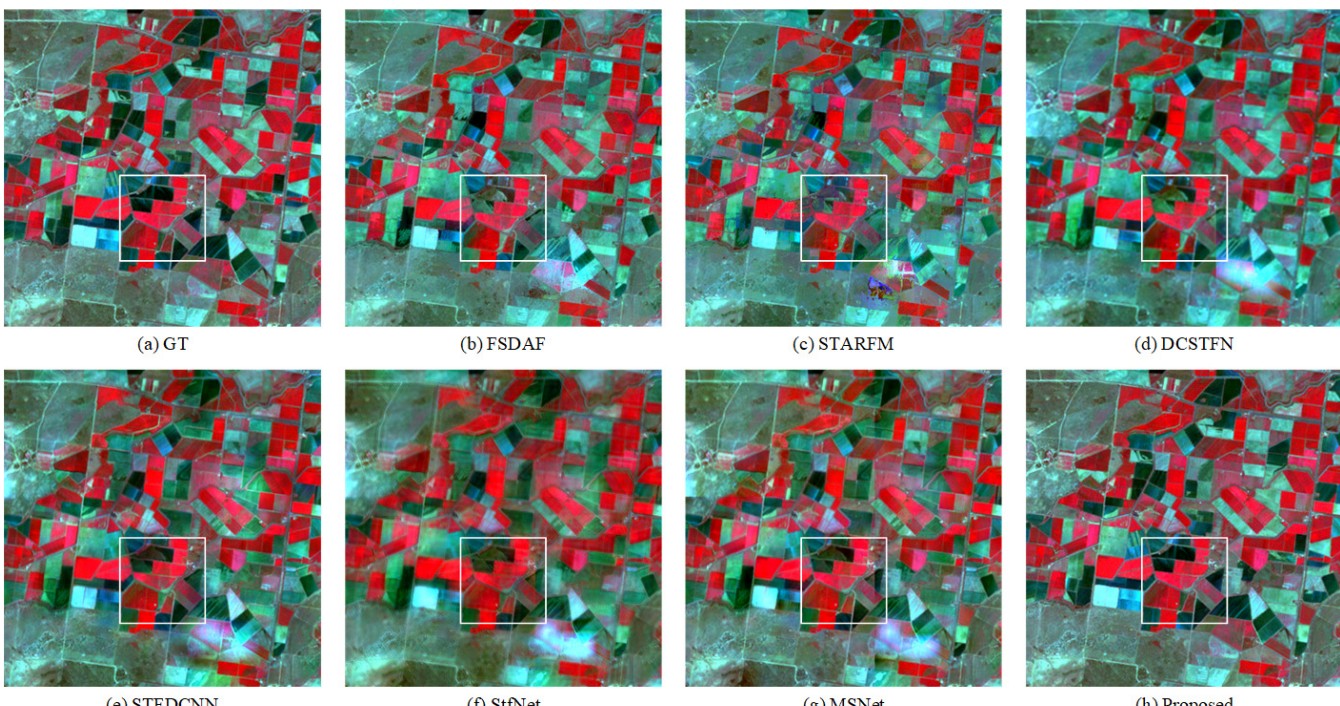

**Figure 8.** Specific prediction results for the target Landsat image (16 October 2001) in CIA dataset. Among them, the white framework is the prominent difference of the results obtained by each method. Comparison methods include FSDAF [11], STARFM [8], DCSTFN [23], STFDCNN [18], StfNet [19], and MSNet [36], which are represented by (**b**–**g**) in the figure respectively. (**a**) represents the ground truth (GT), and (**h**) represents the proposed method.

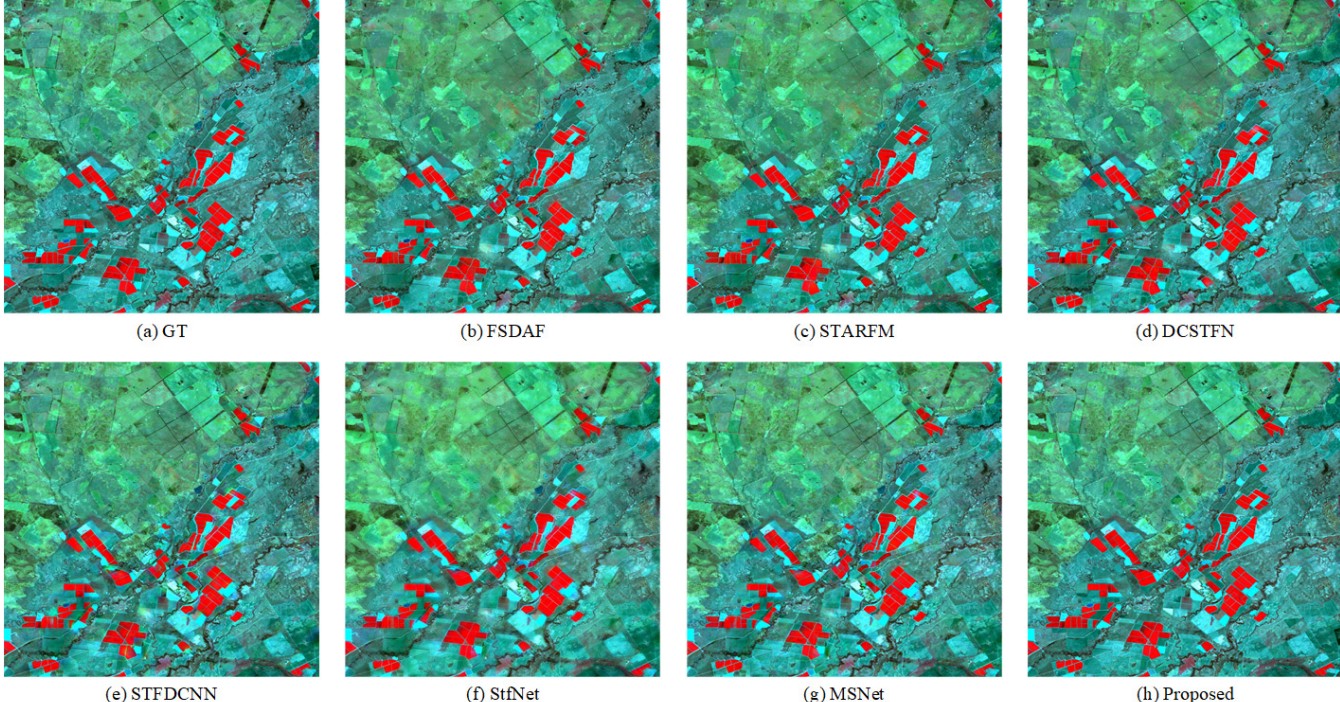

**Figure 9.** Comprehensive prediction results for the target Landsat image (14 February 2005) in LGC dataset. Comparison methods include FSDAF [11], STARFM [8], DCSTFN [23], STFDCNN [18], StfNet [19], and MSNet [36], which are represented by (**b**–**g**) in the figure respectively. (**a**) represents the ground truth (GT), and (**h**) represents the proposed method.

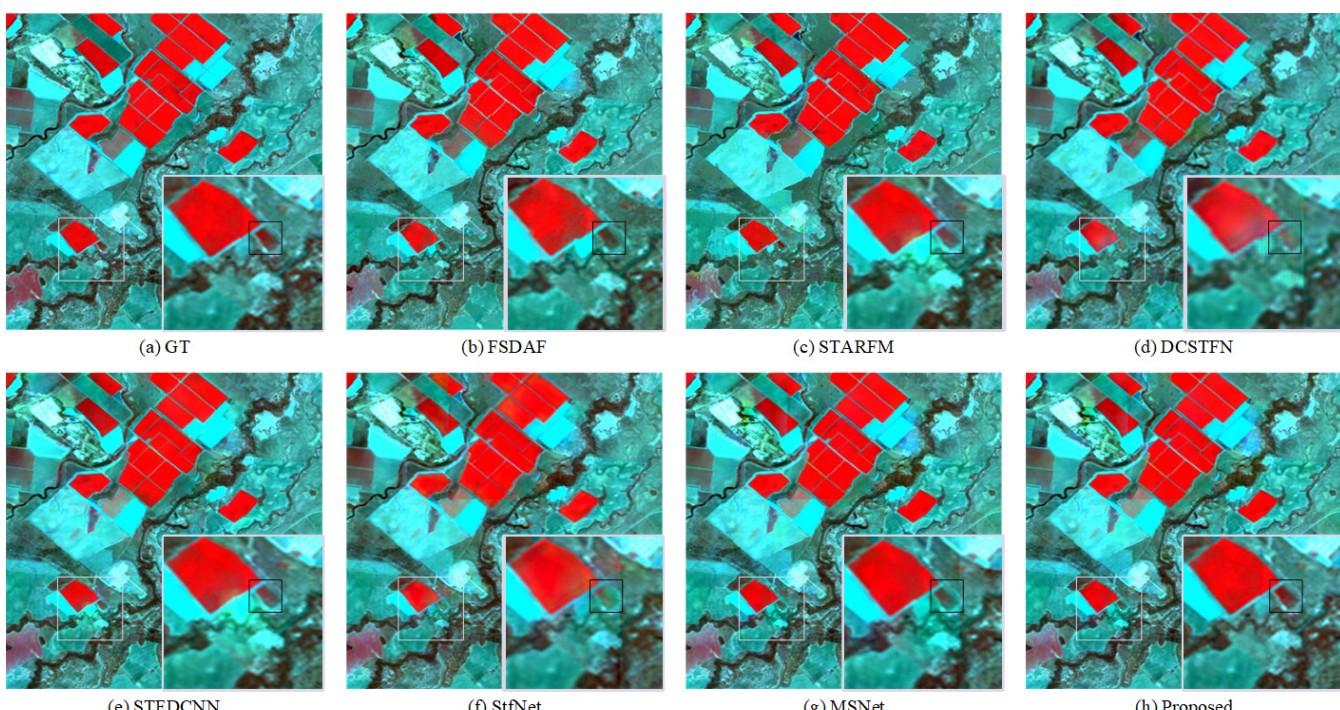

**Figure 10.** Specific prediction results for the target Landsat image (14 February 2005) in LGC dataset. Among them, the grey framework is the prominent difference of the results obtained by each method. Comparison methods include FSDAF [11], STARFM [8], DCSTFN [23], STFDCNN [18], StfNet [19], and MSNet [36], which are represented by (**b**–**g**) in the figure respectively. (**a**) represents the ground truth (GT), and (**h**) represents the proposed method.

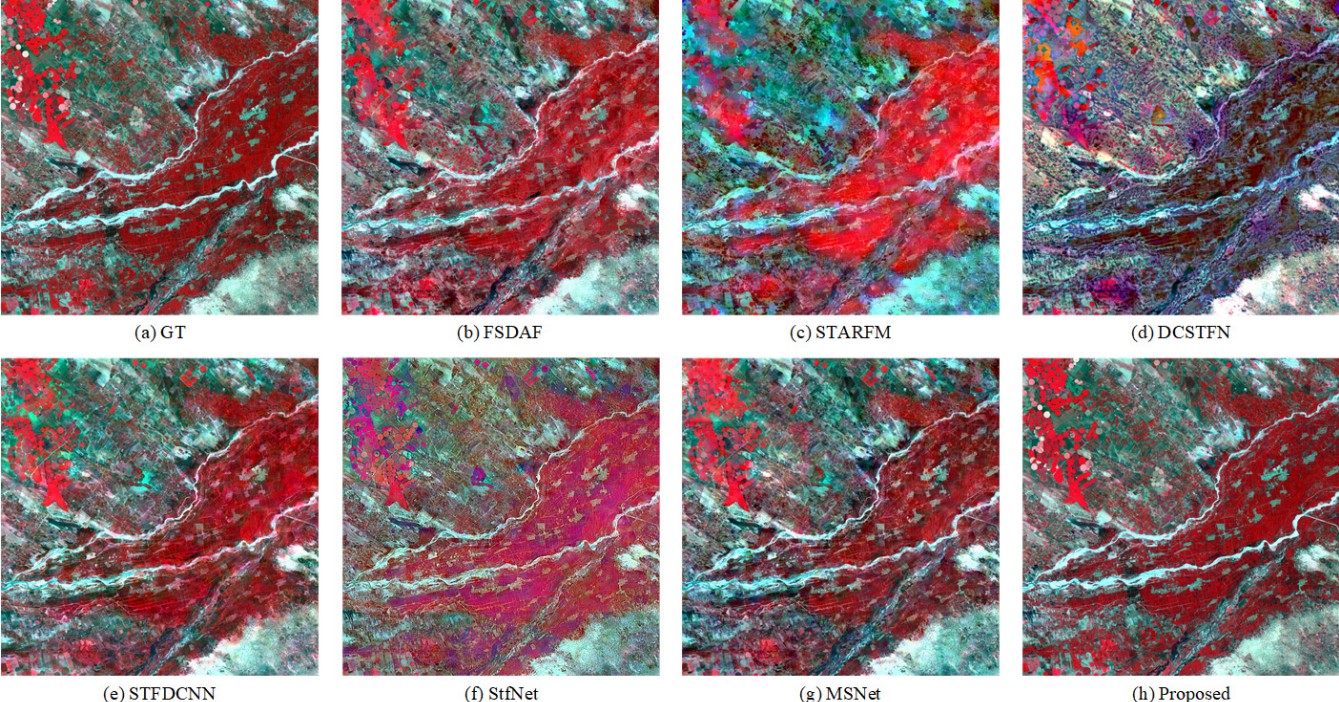

**Figure 11.** Complete prediction results for the target Landsat image (7 July 2015) in AHB dataset. Comparison methods include FSDAF [11], STARFM [8], DCSTFN [23], STFDCNN [18], StfNet [19], and MSNet [36], which are represented by (**b**–**g**) in the figure respectively. (**a**) represents the ground truth (GT), and (**h**) represents the proposed method.

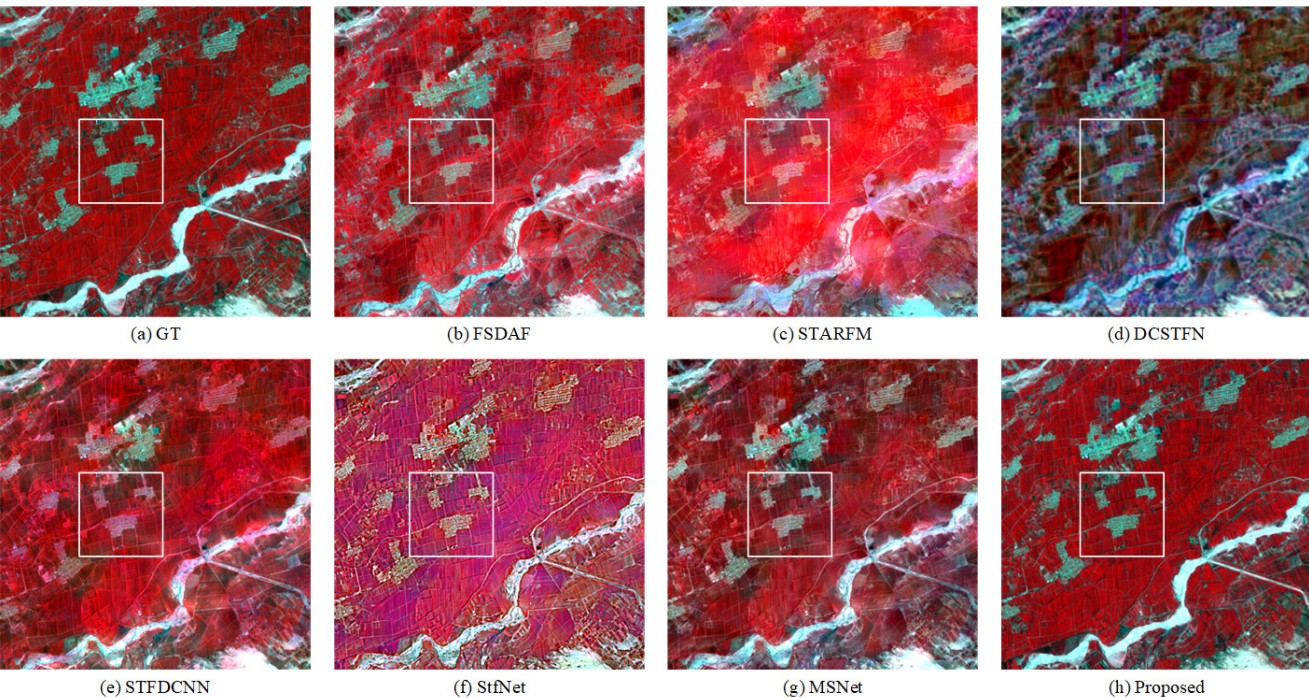

**Figure 12.** First specific prediction results for the target Landsat image (7 July 2015) in AHB dataset. Among them, the white framework is the prominent difference of the results obtained by each method. Comparison methods include FSDAF [11], STARFM [8], DCSTFN [23], STFDCNN [18], StfNet [19], and MSNet [36], which are represented by (**b**–**g**) in the figure respectively. (**a**) represents the ground truth (GT), and (**h**) represents the proposed method.

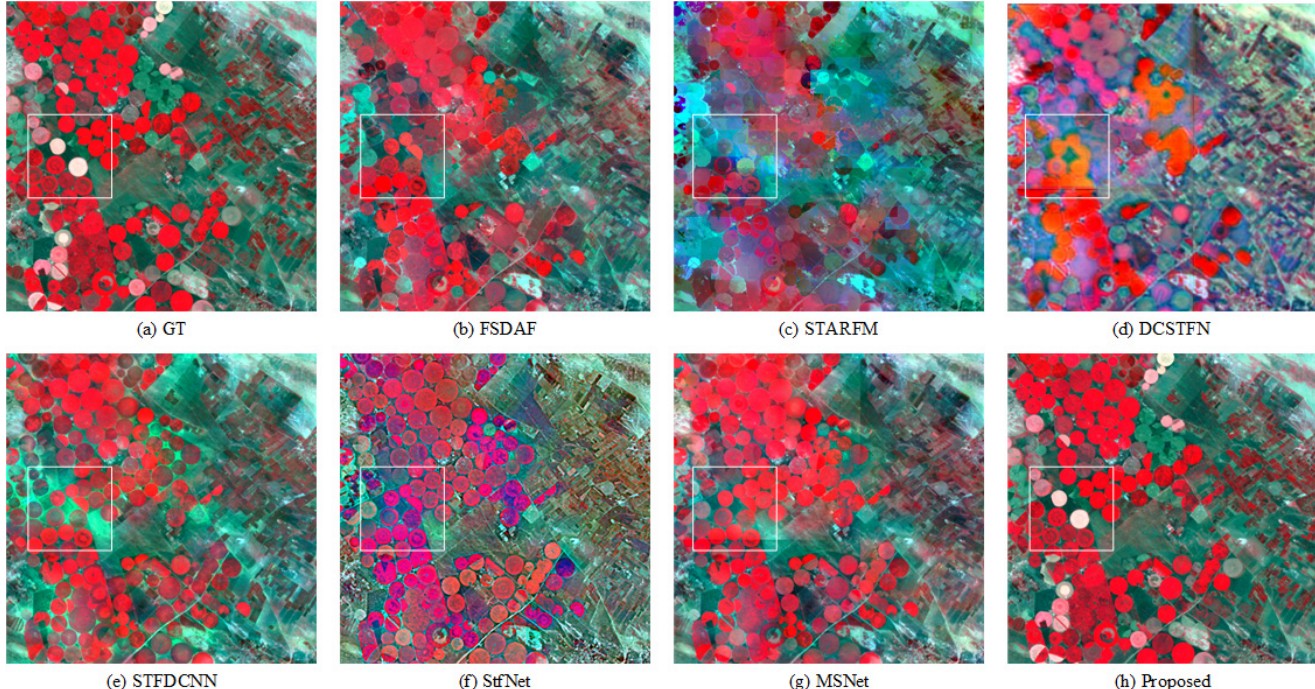

**Figure 13.** Second specific prediction results for the target Landsat image (7 July 2015) in AHB dataset. Among them, the white framework is the prominent difference of the results obtained by each method. Comparison methods include FSDAF [11], STARFM [8], DCSTFN [23], STFDCNN [18], StfNet [19], and MSNet [36], which are represented by (**b**–**g**) in the figure respectively. (**a**) represents the ground truth (GT), and (**h**) represents the proposed method.

Figure 7 shows the overall prediction result on the CIA data set, while Figure 8 shows a cropped part of the prediction result enlarged. Visually, FSDAF, STARFM, and DCSTFN are less accurate than other methods in predicting phenological changes. For example, in the overall results in Figure 7, the black areas of these methods are noticeably less than those contained in GT. The prediction effects in the box in Figure 8 are also quite different. Relatively speaking, the prediction results obtained by the method based on deep learning are better, but the prediction map of StfNet is a bit blurry and the effect is not good. The results of STFDCNN and MSNet are relatively good, but those of our proposed method are better. Thus, Figure 8 shows that the results obtained by the proposed method are closer to the ground truth in terms of clarity and accuracy.

Figure 9 illustrates the overall prediction result on the LGC dataset, while Figure 10 illustrates the cropped and enlarged result of a portion of the prediction. In general, the performance of each algorithm is relatively stable, but there are differences in the specific spectral information and the processing of heterogeneous regions. It can be seen from the black box in the enlarged area in the lower right corner of Figure 10 that the prediction accuracy of the spectral information in DCSTFN and StfNet is lower than other methods, and the other methods have achieved good results, but the effect obtained by the proposed method is closer to the actual value. In addition, the proposed method also predicts the information of curved river channels with high heterogeneity under the black box, which no other method except for FSDAF can. Compared with the proposed method, FSDAF is closer to the real value. The method has achieved good results in spectral information and the processing of heterogeneous regions.

Figure 11 shows the overall prediction result on the AHB data set, while Figures 12 and 13 show some cropped and enlarged results. On the whole, the prediction results of STARFM are not accurate enough in the processing of spectral information, and there is considerable ambiguous spectral information. DCSTFN fails to accurately predict the results, and fails to effectively extract information for datasets with a large number of heterogeneous regions and time information. The results obtained by StfNet are relatively good, such as in the spatial details between rivers, but there is still a large gap between the overall and the real value. In addition, although the prediction results of FSDAF are much better than STARFM in the processing of spectral information, there are still shortcomings compared with the real values. While STFDCNN and MSNet achieve better results, the spatial details and spectral time information are relatively adequate, but the proposed method achieves better results, with the spatial details and spectral information being closer to the real values. Locally, in Figure 12, in a large number of continuous phenological change areas, the proposed method has a noticeable improvement compared with the previous MSNet. Furthermore, compared with other methods, the processing of boundary information is also better, and is closest to the true value. In Figure 13, for the prediction of a large number of circular pasture areas, FSDAF, STARFM, DCSTFN, and StfNet failed at accurate prediction, which must be due to the complex spatial distribution and too much time-varying information on the AHB dataset, which led to the limited learning ability of the model, and the results obtained were not ideal. STFDCNN has achieved good results with the previous MSNet, but there is still insufficient boundary information. The proposed method thus achieves the best prediction effect, in the prediction of phenological change information as well as the boundary processing between circular pastures.

### 3.3.2. Objective Evaluation

Six evaluation indicators are used to objectively evaluate various algorithms and the proposed method. Tables 1–3 present the quantitative evaluation of the prediction results obtained by various methods on three datasets, including global indicators SAM and ERGAS as well as local indicators RMSE, SSIM, PSNR, and CC. Furthermore, the optimal value of each indicator is marked in bold.

**Table 1.** Quantitative assessment of various spatiotemporal fusion methods for CIA dataset.

| Evaluation | Band | Method on CIA | | | | | | |
|---|---|---|---|---|---|---|---|---|
| | | **FSDAF** | **DCSTFN** | **STARFM** | **STFDCNN** | **StfNet** | **MSNet** | **Proposed** |
| SAM | all | 0.23875 | 0.21556 | 0.23556 | 0.21402 | 0.21614 | 0.19209 | **0.00114** |
| ERGAS | all | 3.35044 | 3.07221 | 3.31676 | 3.14461 | 3.00404 | 2.94471 | **0.45234** |
| RMSE | band1 | 0.01365 | 0.01059 | 0.01306 | 0.01076 | 0.00956 | 0.01009 | **0.00051** |
| | band2 | 0.01415 | 0.01256 | 0.01366 | 0.01236 | 0.01271 | 0.01132 | **0.00044** |
| | band3 | 0.02075 | 0.01922 | 0.02055 | 0.01792 | 0.02121 | 0.01724 | **0.00032** |
| | band4 | 0.04619 | 0.04377 | 0.04899 | 0.04100 | 0.05001 | 0.03669 | **0.00079** |
| | band5 | 0.06031 | 0.05655 | 0.06153 | 0.05900 | 0.05302 | 0.04898 | **0.00026** |
| | band6 | 0.05322 | 0.04690 | 0.05278 | 0.05389 | 0.04500 | 0.04325 | **0.00067** |
| | avg | 0.03471 | 0.03160 | 0.03509 | 0.03249 | 0.03192 | 0.02793 | **0.00050** |
| SSIM | band1 | 0.90147 | 0.94678 | 0.91699 | 0.95517 | 0.94190 | 0.95050 | **0.99996** |
| | band2 | 0.91899 | 0.93652 | 0.92325 | 0.93812 | 0.94340 | 0.95149 | **0.99998** |
| | band3 | 0.85786 | 0.88428 | 0.86290 | 0.87329 | 0.89950 | 0.91156 | **0.99999** |
| | band4 | 0.76070 | 0.79776 | 0.74636 | 0.78318 | 0.84868 | 0.86248 | **0.99995** |
| | band5 | 0.66598 | 0.70744 | 0.66011 | 0.72789 | 0.74118 | 0.76460 | **0.99999** |
| | band6 | 0.66168 | 0.72121 | 0.66323 | 0.73555 | 0.74068 | 0.76257 | **0.99997** |
| | avg | 0.79445 | 0.83233 | 0.79548 | 0.83553 | 0.85256 | 0.86720 | **0.99997** |
| PSNR | band1 | 37.29537 | 39.50404 | 37.68327 | 39.36680 | 40.38939 | 39.92510 | **65.81463** |
| | band2 | 36.98507 | 38.01703 | 37.29114 | 38.16128 | 37.91972 | 38.92643 | **67.09016** |
| | band3 | 33.65821 | 34.32276 | 33.74247 | 34.93560 | 33.46842 | 35.27141 | **69.83863** |
| | band4 | 26.70854 | 27.17708 | 26.19858 | 27.74355 | 26.01829 | 28.70879 | **62.06650** |
| | band5 | 24.39249 | 24.95152 | 24.21822 | 24.58366 | 25.51175 | 26.19920 | **71.78578** |
| | band6 | 25.47784 | 26.57641 | 25.55050 | 25.37055 | 26.93525 | 27.28095 | **63.47700** |
| | avg | 30.75292 | 31.75814 | 30.78070 | 31.69357 | 31.70714 | 32.71865 | **66.67879** |
| CC | band1 | 0.80138 | 0.79672 | 0.79845 | 0.84521 | 0.83428 | 0.84448 | **0.99951** |
| | band2 | 0.79873 | 0.81009 | 0.79319 | 0.83720 | 0.83156 | 0.84929 | **0.99978** |
| | band3 | 0.83290 | 0.84688 | 0.82554 | 0.87373 | 0.87264 | 0.87787 | **0.99996** |
| | band4 | 0.88511 | 0.89683 | 0.86697 | 0.91181 | 0.90546 | 0.92743 | **0.99997** |
| | band5 | 0.76395 | 0.79363 | 0.74894 | 0.78783 | 0.84732 | 0.84784 | **0.99999** |
| | band6 | 0.76036 | 0.80739 | 0.75144 | 0.76502 | 0.84588 | 0.83826 | **0.99996** |
| | avg | 0.80707 | 0.82526 | 0.79742 | 0.83680 | 0.85619 | 0.86420 | **0.99986** |

**Table 2.** Quantitative assessment of various spatiotemporal fusion methods for LGC dataset.

| Evaluation | Band | Method on LGC | | | | | | |
|---|---|---|---|---|---|---|---|---|
| | | **FSDAF** | **DCSTFN** | **STARFM** | **STFDCNN** | **StfNet** | **MSNet** | **Proposed** |
| SAM | all | 0.08411 | 0.08354 | 0.08601 | 0.06792 | 0.09284 | 0.06335 | **0.00035** |
| ERGAS | all | 1.93861 | 1.91167 | 1.92273 | 1.80392 | 2.03970 | 1.68639 | **0.13248** |
| RMSE | band1 | 0.00763 | 0.00763 | 0.00729 | 0.00719 | 0.00824 | 0.00585 | **0.00006** |
| | band2 | 0.00913 | 0.00870 | 0.00907 | 0.00843 | 0.01167 | 0.00712 | **0.00006** |
| | band3 | 0.01279 | 0.01258 | 0.01256 | 0.01151 | 0.01353 | 0.00969 | **0.00006** |
| | band4 | 0.02383 | 0.02332 | 0.02295 | 0.02102 | 0.02971 | 0.01864 | **0.00006** |
| | band5 | 0.02830 | 0.02679 | 0.02607 | 0.02251 | 0.02284 | 0.02159 | **0.00006** |
| | band6 | 0.02197 | 0.02072 | 0.02181 | 0.01673 | 0.02054 | 0.01425 | **0.00006** |
| | avg | 0.01727 | 0.01662 | 0.01662 | 0.01457 | 0.01775 | 0.01286 | **0.00006** |
| SSIM | band1 | 0.97422 | 0.97455 | 0.97355 | 0.98460 | 0.97464 | 0.98558 | **0.99999** |
| | band2 | 0.96698 | 0.96918 | 0.96495 | 0.98209 | 0.96062 | 0.98031 | **0.99999** |
| | band3 | 0.94456 | 0.94632 | 0.94152 | 0.97475 | 0.94162 | 0.96954 | **0.99999** |
| | band4 | 0.92411 | 0.93283 | 0.91759 | 0.96417 | 0.91455 | 0.96393 | **0.99999** |
| | band5 | 0.89418 | 0.90416 | 0.88558 | 0.95539 | 0.91215 | 0.95239 | **0.99999** |
| | band6 | 0.88485 | 0.90337 | 0.87789 | 0.95259 | 0.90154 | 0.95087 | **0.99999** |
| | avg | 0.93148 | 0.93840 | 0.92684 | 0.96893 | 0.93419 | 0.96710 | **0.99999** |

**Table 2.** *Cont.*

| Evaluation | Band | Method on LGC | | | | | | |
|---|---|---|---|---|---|---|---|---|
| | | **FSDAF** | **DCSTFN** | **STARFM** | **STFDCNN** | **StfNet** | **MSNet** | **Proposed** |
| PSNR | band1 | 42.35483 | 42.34734 | 42.73997 | 42.86245 | 41.68016 | 44.65345 | **84.25491** |
| | band2 | 40.79034 | 41.20550 | 40.85222 | 41.48586 | 38.65611 | 42.95050 | **85.06363** |
| | band3 | 37.86428 | 38.00486 | 38.02099 | 38.77733 | 37.37629 | 40.27059 | **83.84853** |
| | band4 | 32.45760 | 32.64622 | 32.78532 | 33.54859 | 30.54336 | 34.59058 | **84.47622** |
| | band5 | 30.96416 | 31.44212 | 31.67671 | 32.95179 | 32.82613 | 33.31671 | **84.01371** |
| | band6 | 33.16535 | 33.67016 | 33.22812 | 35.52920 | 33.74927 | 36.92082 | **84.00084** |
| | avg | 36.26610 | 36.55270 | 36.55056 | 37.52587 | 35.80522 | 38.78378 | **84.27631** |
| CC | band1 | 0.93627 | 0.92666 | 0.92935 | 0.94611 | 0.94664 | 0.96138 | **0.99999** |
| | band2 | 0.93186 | 0.93379 | 0.92880 | 0.94530 | 0.93566 | 0.95800 | **0.99999** |
| | band3 | 0.93549 | 0.93512 | 0.93516 | 0.95262 | 0.95539 | 0.96499 | **0.99999** |
| | band4 | 0.96360 | 0.96585 | 0.96287 | 0.97181 | 0.96125 | 0.97591 | **0.99999** |
| | band5 | 0.95527 | 0.95492 | 0.95222 | 0.97545 | 0.97048 | 0.97890 | **0.99999** |
| | band6 | 0.95313 | 0.95738 | 0.95214 | 0.97285 | 0.97164 | 0.97924 | **0.99999** |
| | avg | 0.94594 | 0.94562 | 0.94342 | 0.96069 | 0.95684 | 0.96974 | **0.99999** |

**Table 3.** Quantitative assessment of various spatiotemporal fusion methods for AHB dataset.

| Evaluation | Band | Method on AHB | | | | | | |
|---|---|---|---|---|---|---|---|---|
| | | **FSDAF** | **DCSTFN** | **STARFM** | **STFDCNN** | **StfNet** | **MSNet** | **Proposed** |
| SAM | all | 0.16991 | 0.23877 | 0.29277 | 0.18583 | 0.25117 | 0.14677 | **0.01297** |
| ERGAS | all | 2.80156 | 4.03380 | 4.46147 | 4.25224 | 3.86535 | 2.90661 | **0.81967** |
| RMSE | band1 | 0.00039 | 0.00081 | 0.00251 | 0.00096 | 0.00112 | 0.00047 | **0.00007** |
| | band2 | 0.00044 | 0.00215 | 0.00235 | 0.00092 | 0.00081 | 0.00051 | **0.00007** |
| | band3 | 0.00067 | 0.00363 | 0.00358 | 0.00117 | 0.00118 | 0.00064 | **0.00007** |
| | band4 | 0.00109 | 0.00187 | 0.00590 | 0.00124 | 0.00201 | 0.00103 | **0.00006** |
| | band5 | 0.00126 | 0.00208 | 0.00408 | 0.00183 | 0.00177 | 0.00122 | **0.00006** |
| | band6 | 0.00136 | 0.00225 | 0.00263 | 0.00200 | 0.00198 | 0.00126 | **0.00007** |
| | avg | 0.00087 | 0.00213 | 0.00351 | 0.00135 | 0.00148 | 0.00085 | **0.00006** |
| SSIM | band1 | 0.99895 | 0.99459 | 0.96538 | 0.99205 | 0.98927 | 0.99822 | **0.99998** |
| | band2 | 0.99877 | 0.96845 | 0.96977 | 0.99293 | 0.99500 | 0.99805 | **0.99998** |
| | band3 | 0.99741 | 0.91914 | 0.93438 | 0.98947 | 0.98965 | 0.99740 | **0.99998** |
| | band4 | 0.99616 | 0.98506 | 0.92038 | 0.99419 | 0.98248 | 0.99631 | **0.99999** |
| | band5 | 0.99382 | 0.98085 | 0.94190 | 0.98371 | 0.98464 | 0.99388 | **0.99999** |
| | band6 | 0.99129 | 0.97145 | 0.96825 | 0.97625 | 0.97636 | 0.99226 | **0.99998** |
| | avg | 0.99607 | 0.96992 | 0.95001 | 0.98810 | 0.98623 | 0.99602 | **0.99998** |
| PSNR | band1 | 68.18177 | 61.87013 | 52.01008 | 60.34502 | 59.00582 | 66.48249 | **83.62824** |
| | band2 | 67.04371 | 53.35105 | 52.56484 | 60.68929 | 61.8316 | 65.80339 | **83.61930** |
| | band3 | 63.49068 | 48.79810 | 48.93197 | 58.63694 | 58.55977 | 63.88021 | **83.56309** |
| | band4 | 59.22553 | 54.57435 | 44.58211 | 58.13169 | 53.95486 | 59.77506 | **84.23956** |
| | band5 | 58.02282 | 53.65469 | 47.79106 | 54.74701 | 55.05539 | 58.28599 | **83.87554** |
| | band6 | 57.35352 | 52.93719 | 51.60634 | 53.96601 | 54.06602 | 58.02322 | **83.56037** |
| | avg | 62.21967 | 54.19759 | 49.58107 | 57.75266 | 57.07891 | 62.04173 | **83.74768** |
| CC | band1 | 0.84000 | 0.78227 | 0.71181 | 0.80368 | 0.49726 | 0.86845 | **0.99570** |
| | band2 | 0.85657 | 0.76351 | 0.74545 | 0.86845 | 0.38062 | 0.89114 | **0.99795** |
| | band3 | 0.84979 | 0.79147 | 0.81230 | 0.83576 | 0.27147 | 0.88345 | **0.99918** |
| | band4 | 0.53986 | 0.40161 | 0.34009 | 0.58944 | 0.37556 | 0.60303 | **0.99893** |
| | band5 | 0.79576 | 0.52206 | 0.76553 | 0.83580 | 0.62926 | 0.85320 | **0.99972** |
| | band6 | 0.80288 | 0.47565 | 0.76492 | 0.80338 | 0.61085 | 0.85154 | **0.99975** |
| | avg | 0.78081 | 0.62276 | 0.69002 | 0.78942 | 0.46083 | 0.82514 | **0.99854** |

Tables 1–3 present the quantitative evaluation results of several existing fusion methods and the proposed method on the CIA, LGC, and AHB datasets, respectively. In each

table, it can be seen that the proposed method achieves the optimal value on the global indicators and all local indicators.

## 4. Discussion

Through the experiments, it can be seen that whether it is on the CIA dataset with phenological changes in regular areas or on the AHB dataset with phenological changes with a large number of irregular areas and a large number of heterogeneous areas, our proposed method is better at prediction. Similarly, for LGC datasets, which are mainly land cover-type changes, the proposed method is better at prediction than traditional methods and other deep learning-based methods in the processing of temporal information and high-frequency spatial details. The time information and high-frequency file texture information are processed more appropriately because of the combination of ITE and dilated convolution in EMSNet. More importantly, the refined ITE can further expand the range of learning in the remote sensing field, and can fully extract the spatiotemporal information contained in the input image.

It is worth noting that for datasets with different amounts of data and different characteristics, the depth of the improved transformer encoder (ITE) should also be different to better fit the datasets. Table 4 lists the average evaluation values of the prediction results obtained without the ITE and with the ITE with different depths, where the optimal value is shown in bold. The depth being 0 indicates that the ITE has not been introduced. It can be seen that when the depth is not introduced, the experimental results are relatively poor. As the depth changes, the results obtained vary. The best experimental results are obtained when the depth of the CIA dataset is 20, the depth of the LGC dataset is 5, and the depth of the AHB dataset is 20.

**Table 4.** Average evaluation values of ITEs of various depths on the three datasets.

| Database | Depth | SAM | ERGAS | RMSE | SSIM | PSNR | CC |
|----------|-------|-----|-------|------|------|------|-----|
| CIA | 0 | 0.223768 | 3.144353 | 0.032796 | 0.844214 | 31.477961 | 0.819219 |
| | 5 | 0.001597 | 0.530676 | 0.000550 | 0.999948 | 67.018571 | 0.999620 |
| | 10 | 0.001182 | 0.473233 | **0.000473** | 0.999971 | **68.362024** | 0.999807 |
| | 15 | 0.001394 | 0.509978 | 0.000639 | 0.999960 | 64.859474 | 0.999776 |
| | **20** | **0.001142** | **0.452341** | 0.000499 | **0.999974** | 66.678786 | **0.999863** |
| LGC | 0 | 0.082166 | 1.939385 | 0.016704 | 0.943749 | 36.315476 | 0.948030 |
| | **5** | **0.000352** | **0.132476** | **0.000061** | **0.9999982** | **84.276309** | **0.9999989** |
| | 10 | 0.000367 | 0.139728 | 0.000069 | 0.9999979 | 83.319692 | 0.9999987 |
| | 15 | 0.000378 | 0.153687 | 0.000092 | 0.9999976 | 81.181723 | 0.999998 |
| | 20 | 0.000638 | 0.287639 | 0.000476 | 0.999885 | 77.511986 | 0.999900 |
| AHB | 0 | 0.082166 | 1.939385 | 0.016704 | 0.943749 | 36.315476 | 0.748201 |
| | 5 | 0.013112 | 0.826490 | 0.000066 | 0.999982 | 83.686718 | 0.998556 |
| | 10 | 0.013106 | 0.825792 | 0.000066 | 0.999982 | 83.680289 | **0.998557** |
| | 15 | 0.013102 | 0.828641 | 0.000066 | 0.999982 | 83.625675 | 0.998539 |
| | **20** | **0.012967** | **0.819673** | **0.000065** | **0.999983** | **83.747684** | 0.998540 |

The bold in the table indicates the optimal value at different ITE depths.

In addition, the difference between the original linear projection method of the transformer encoder and the improved convolution projection method was also determined. Table 5 lists the global indicators and average evaluation values of the prediction results obtained under various projection methods, where the optimal value is shown in bold. It can be seen that on the three datasets, the convolutional projection method is selected, and the ITE after position encoding is removed achieves better results.

**Table 5.** Average evaluation values of ICTEs of various project methods on the three datasets.

| Database | Project Method | SAM | ERGAS | RMSE | SSIM | PSNR | CC |
|---|---|---|---|---|---|---|---|
| CIA | line | 0.001142 | 0.452660 | 0.000500 | 0.999966 | 65.565960 | 0.999462 |
| | **conv** | **0.001141** | **0.452341** | **0.000499** | **0.999974** | **66.678786** | **0.999863** |
| LGC | line | 0.000352 | 0.133565 | 0.000070 | 0.999990 | 82.659593 | 0.999984 |
| | **conv** | **0.000351** | **0.132476** | **0.000061** | **0.999998** | **84.276309** | **0.999999** |
| AHB | line | 0.013024 | 0.823650 | 0.000066 | 0.999896 | 81.265960 | 0.990570 |
| | **conv** | **0.012967** | **0.819673** | **0.000065** | **0.999983** | **83.747684** | **0.998540** |

Furthermore, the last six layers of the network for extracting time information in Figure 3 include three layers of dilated convolution and three layers of ReLU. This paper also conducts a comparative experiment on the three layers of dilated convolution operations. Table 6 lists the different result evaluations obtained when using convolution and dilated convolution. Among them, "conv" in the difference column means to replace the above-mentioned three layers of dilated convolution with three layers of convolution; "conv_dia" means that the above-mentioned three layers of dilated convolution remain unchanged, and "conv&conv_dia" means that the abovementioned three layers of dilated convolution are replaced by a three-layer alternating operation of convolution, dilated convolution and convolution. It can be seen that when the subsequent operations of extracting time information are all dilated convolutions, the implementation effect is better.

**Table 6.** Average evaluation values of various convolution operations on the three datasets.

| Database | Difference | SAM | ERGAS | RMSE | SSIM | PSNR | CC |
|---|---|---|---|---|---|---|---|
| CIA | conv | 0.001491 | 0.549008 | 0.000593 | 0.999953 | 66.089266 | 0.999672 |
| | **conv_dia** | **0.001142** | **0.452341** | **0.000499** | **0.999974** | **66.678786** | **0.999863** |
| | conv&conv_dia | 0.101984 | 2.197340 | 0.015059 | 0.943943 | 37.726046 | 0.963180 |
| LGC | conv | 0.000365 | 0.136848 | 0.000064 | 0.9999980 | 83.841604 | 0.9999988 |
| | **conv_dia** | **0.000352** | **0.132476** | **0.000061** | **0.9999982** | **84.276309** | **0.9999989** |
| | conv&conv_dia | 0.050764 | 1.532304 | 0.010584 | 0.975687 | 40.476035 | 0.980252 |
| AHB | conv | 0.012998 | 0.826653 | 0.000066 | 0.999982 | 83.658194 | 0.998290 |
| | **conv_dia** | **0.012967** | **0.819673** | **0.000065** | **0.999983** | **83.747684** | **0.998540** |
| | conv&conv_dia | 0.091340 | 2.057066 | 0.000496 | 0.998751 | 66.825864 | 0.921079 |

Although the proposed method has achieved good results, there are issues worthy of further exploration. First, in order to fully expand the learnable range of the ITE, the original input of a larger MODIS image has been used. Although dilated convolution is used to reduce the number of parameters, compared with MSNet, the number of parameters in this study is quite high. Table 7 presents the fusion model of deep learning and the number of parameters that the proposed method needs to learn. It can be seen that the proposed method needs the largest number of parameters, which means that compared with other methods, it requires more training time and equipment with larger memory during training. Considering the cost of learning, a way to obtain better results with a smaller model is a direction worthy of future research. Second, the refined ITE shows very good performance, but further improvements to adapt it to remote sensing spatiotemporal fusion can be researched in future. Furthermore, improving the fusion effect while avoiding the fusion strategy introduced by noise is also worthy of further study.

**Table 7.** Number of parameters for different deep learning methods.

| Method | DCSTFN | STFDCNN | StfNet | MSNet | | Proposed | |
|---|---|---|---|---|---|---|---|
| | | | | depth = 5 | 521,064 | depth = 5 | 3,673,617 |
| **Parameter** | 445,889 | 114,562 | 36,866 | depth = 10 | 978,764 | depth = 10 | 7,329,217 |
| | | | | depth = 20 | 1,894,164 | depth = 20 | 14,640,417 |

## 5. Conclusions

In this study, the effectiveness of EMSNet in three research areas with diverse characteristics is evaluated. Its performance enhancement is found to be mainly because of the following reasons:

1. The projection method of the original transformer encoder is improved to adapt to the fusion of remote sensing space and time, which further expands the learning range of the improved transformer encoder, effectively learns the connection between the local and the global information in the remote sensing image, and uses its own attention mechanism to fully extract the spatiotemporal information in remote sensing images.
2. Dilated convolution is used to expand the receptive field to adapt to the original input of larger size, while keeping the number of learned parameters unchanged, effectively extracting time information and balancing the increase in parameters brought about by the improved transformer encoder.
3. A unique residual structure and a differentiated weight fusion method are used to supplement the lost information and reduce the introduction of noise in the fusion process.

Experiments show that on the CIA and AHB datasets with noteworthy phenological changes and the LGC dataset with mainly land cover-type changes, EMSNet is better than other models using three and five original images for fusion and gives more stable prediction results on each dataset. Although EMSNet achieves good results, there are still many areas worth further research in the future. First, the application of transformer-related structures in the field of remote sensing spatiotemporal fusion will be further studied. Second, compared with other methods, the method proposed in this paper needs to learn significantly more parameters. How to achieve better fusion effect with smaller model and lower learning cost is also a focus of future research. Third, although the three datasets used in this paper cover a variety of phenological changes and land-cover changes, there are still regional types that are not included. For example, datasets containing changes in urban areas will also be discussed in the future.

**Author Contributions:** Data curation, W.L.; formal analysis, W.L.; methodology, W.L. and D.C.; validation, D.C.; visualization, D.C.; writing—original draft, D.C.; writing—review and editing, D.C. and M.X. All authors have read and agreed to the published version of the manuscript.

**Funding:** This work was supported by the National Natural Science Foundation of China (61972060, U1713213, and 62027827), National Key Research and Development Program of China (2019YFE0110800), and Natural Science Foundation of Chongqing (cstc2020jcyj-zdxmX0025, cstc2019cxcyljrc-td0270).

**Data Availability Statement:** Data sharing is not applicable to this article.

**Acknowledgments:** The authors would like to thank all of the reviewers for their valuable contributions to our work.

**Conflicts of Interest:** The authors declare no conflict of interest.

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
