# Peer review of "Enhanced Multi-Stream Remote Sensing Spatiotemporal Fusion Network Based on Transformer and Dilated Convolution"

_remotesensing, doi:10.3390/rs14184544_

Round 1

Reviewer 1 Report (Previous Reviewer 1)

In this paper, the authors made serious revisions, but the revision did not improve the novelty of the proposed method. The paper can not be published due to the limited novelty. 

Author Response

Reviewer 2 Report (New Reviewer)

Experiments by the author show that on the CIA and AHB datasets with notable phenological changes, as well as the LGC dataset with primarily land cover type changes, EMSNet outperforms other models using three and five original images for fusion and provides more stable prediction results on each dataset. The results of the manuscript are reproducible based on the information provided in the methods section. The manuscript is well written and should pique the readers' interest. The only criticism is that the conclusion should include more information about their future work.

Author Response

Reviewer 3 Report (New Reviewer)

REMOTE SENSING (MDPI)

---------------------

"Enhanced Multi-Stream Remote Sensing Spatiotemporal Fusion Network Based on Convolutional Transformer and Dilated Convolution"

by Weisheng Li, Dongwen Cao, and Minghao Xiang

GENERAL COMMENTS

----------------

1. The submitted paper presents a novel DNN solution of the following interesting problem:
Having the sequence of the satellite, multi-spectral images made for the given earth area, being acquired with the low temporal resolution (say once per 10 days) and the high spatial resolution (say 1200x1200 pixels), improve the spatial resolution of another image sequence acquired for the same area with the higher temporal resolution (say two times daily) but with the lower spatial resolution (say 75x75 pixels).

2. By a first glance it seems that the problem is ill-posed. How can we expect good prediction of details in the given area patch having only its average signal in the measured spectrum band and details five days before? By the second thought possibility is explained by nature of the measurements. Actually we are not predicting any motion for earth fragments. We predict the change of their spectral content.

3. The proposed DNN architecture for solving the above problem is based on two neural components (say C1 and C2) with outputs being aggregated by the learned weights. The component C1 adds to the detailed image L1 the enhanced (E1) difference M2-M1 of target rough image M2 and source rough image M1 (actually it computes L1+E1(M2-M1)), while the component C2 makes the enhancement E2 of the sum L2+(M2-M1), i.e. it computes E2(L2+M2-M1).

The component C1 consists of four convolution layers, interleaved by three ReLU activations. The dilation factors for those convolution layers are

1,2,3,4, respectively.

More elaborated is the component C2 which begins from the convolution layer (dilation=1), followed by a cascade of 9-head transformers with self attention layers only. Standard ReLU activations are replaced in those transformers by GELU activations.

4. Authors have experimented with 5,10, and 20 transformers in the cascade.

The loss function comparing L2 with its prediction is the Huber loss while the six computed metrics are: SAM, RMSE, ERGAS, SSIM, PSNR, CC.

5. We can say that the results of the above metrics are "more than perfect". Just to mention that SSIM equals to 1 with accuracy of four decimal digits (= 0.9999...).

6. My concern is only on the results of SAM (Spectral Angle Mapper) metrics. For instance, it is not possible to get average angle equal to about 0.5 radians between almost equal spectral vectors, unless the units are confused, i.e. instead of radians we use degrees.

7. Beside of the perfect results, the problem itself, its background, the solution including the DNN architecture, the comprehensive analysis of the results, and the ablation study - all details are, with few exceptions well presented. Therefore I recommend this paper for publication in the "MDPI Remote Sensing" journal. Of course, my recommendation will be fully valid after the minor revisions implemented by the authors. They are explained below, in my detailed comments.

DETAILED COMMENTS

-----------------

1. It seems that your convolutional transformer ICTE is not exactly the same as described in the referenced paper "Cvt: Introducing convolutions to vision transformers". In Figure 2 you make convolutional projection only once. In Cvt they make it after each transformer-encoder. If this is the case, your name "Improved CTE" is problematic and also the name Convolutional Transformer used in the title is confusing like the "Forward unit" in place of MLP used in Figure 2.

In case when your change for the classical transformer (like in MSNet) beside GELU activation is skipping the positional code and preceding the whole transformer by the convolution layer you should not use the name Convolutional Transformer as it is not. The classical transformer gets only new features at its input which (in your case) are computed by the single input convolutional layer (like in DETR detector which is also based on ViT)!

Pls. make this all clear:

(a) either refine the Figure 2 (like in Cvt)

(b) or use the established terminology in your paper by replacing the name "Convolutional Transformer" by the name "Transformer" only.

2. Yes, you had explained the six metrics SAM,...,CC in your published paper on MSnet but why the reader have to look for them. Pls. copy the formulas to the current paper. BTW, check units in the definition of SAM metrics (see my general comment no. 6).

3. In Figure 3 if you change the color of case dilation=1 from cyan to blue then we get matching of colors with the architecture. Just a suggestion.

4. In the formula 3 you use PHI(x) term which suggests that ReLU activation includes processing of layer defined by formula (2). Just change (3) to: ReLu(x) = max(0,x).

5. In formula (6), the Huber loss is defined using scalar operations (abs, square) on two tensors. This confusion can be removed if you precede them by summation over pixels and add to the formula the pixels indexes.

6. In Table 6, the column labeled by "difference" is not clear. Those "last six layer of convolution blocks with dilations" are not clear, as well. I understand that your C1 component (see my general comment no. 3 for C1,C2 notation) there are only four convolutions shown in Figures 1 and 3. Moreover, in C2 there is no dilations at all. Pls. make it clear!

7. We observe significant increase of weights between your MSNet solution and the proposed EMSNet architecture. What about average time increase in the inference stage.

Round 2

Reviewer 1 Report (Previous Reviewer 1)

In this paper, the author revised the paragraphs of the manuscript. However, the paper still can NOT meet the requirement of a journal paper. 

1. I understand that it is not easy to obtain these data, which is also an important contribution of this paper. However, this is not the determinative factor to accept a paper. 

2. The novelty of the proposed method is quite limited. The spatiotemporal fusion is not a new idea in remote sensing community. Furthermore, the dilated convolution neural network has been widely applied to remote sensing image classification and target recognition. 

3. The author did not improve the presentation of the experiment results, which has been proposed in the first round of review. 

Considering all the above, the paper can NOT be accepted in the current form, unless the proposed method is improved. 

This manuscript is a resubmission of an earlier submission. The following is a list of the peer review reports and author responses from that submission.

Round 1

Reviewer 1 Report

In this paper, the author proposed a fusion network for remote sensing image spatiotemporal fusion. I am afraid this paper can not be accepted. 

1. The novelty of this paper is quite limited.

2. The title is too general to cover the main idea of the proposed method. 

3. The showcases of tables and figures in this paper are compulsory to improve.  

Reviewer 2 Report

The paper is well developed with exciting results. However some adjustments should be done, in the abstract, it is better to mention the improvement that the proposed model in percentage makes also adding scale to maps is necessary since the input datasets have different scales. 

Reviewer 3 Report

File attached
